# Riemannian SVRG: Fast Stochastic Optimization on Riemannian Manifolds

**Hongyi Zhang**
hongyiz@mit.edu
MIT

**Sashank J. Reddi**
sjakkamr@cs.cmu.edu
Carnegie Mellon University

**Suvrit Sra**
suvrit@mit.edu
MIT

## Abstract

We study optimization of finite sums of *geodesically* smooth functions on Riemannian manifolds. Although variance reduction techniques for optimizing finite-sums have witnessed tremendous attention in the recent years, existing work is limited to vector space problems. We introduce *Riemannian SVRG* (RSVRG), a new variance reduced Riemannian optimization method. We analyze RSVRG for both geodesically *convex* and *nonconvex* (smooth) functions. Our analysis reveals that RSVRG inherits advantages of the usual SVRG method, but with factors depending on curvature of the manifold that influence its convergence. To our knowledge, RSVRG is the first *provably fast* stochastic Riemannian method. Moreover, our paper presents the first non-asymptotic complexity analysis (novel even for the batch setting) for nonconvex Riemannian optimization. Our results have several implications; for instance, they offer a Riemannian perspective on variance reduced PCA, which promises a short, transparent convergence analysis.

## 1 Introduction

We study the following rich class of (possibly nonconvex) finite-sum optimization problems:

$$\min_{x \in \mathcal{X} \subset \mathcal{M}} f(x) \triangleq \frac{1}{n} \sum_{i=1}^{n} f_i(x), \tag{1}$$

where $(\mathcal{M}, \mathfrak{g})$ is a Riemannian manifold with the Riemannian metric $\mathfrak{g}$, and $\mathcal{X} \subset \mathcal{M}$ is a geodesically convex set. We assume that each $f_i : \mathcal{M} \to \mathbb{R}$ is geodesically $L$-smooth (see §2). Problem (1) generalizes the fundamental machine learning problem of empirical risk minimization, which is usually cast in vector spaces, to a Riemannian setting. It also includes as special cases important problems such as principal component analysis (PCA), independent component analysis (ICA), dictionary learning, mixture modeling, among others (see e.g., the related work section).

The Euclidean version of (1) where $\mathcal{M} = \mathbb{R}^d$ and $\mathfrak{g}$ is the Euclidean inner-product has been the subject of intense algorithmic development in machine learning and optimization, starting with the classical work of Robbins and Monro [26] to the recent spate of work on variance reduction [10; 18; 20; 25; 28]. However, when $(\mathcal{M}, \mathfrak{g})$ is a nonlinear Riemannian manifold, much less is known beyond [7; 38].

When solving problems with manifold constraints, one common approach is to alternate between optimizing in the ambient Euclidean space and "projecting" onto the manifold. For example, two well-known methods to compute the leading eigenvector of symmetric matrices, power iteration and Oja's algorithm [23], are in essence projected gradient and projected stochastic gradient algorithms. For certain manifolds (e.g., positive definite matrices), projections can be quite expensive to compute.

An effective alternative is to use *Riemannian optimization*[1], which directly operates on the manifold in question. This mode of operation allows Riemannian optimization to view the constrained optimization problem (1) as an unconstrained problem on a manifold, and thus, to be "projection-free." More important is its conceptual value: viewing a problem through the Riemannian lens, one can discover insights into problem geometry, which can translate into better optimization algorithms.

Although the Riemannian approach is appealing, our knowledge of it is fairly limited. In particular, there is little analysis about its global complexity (a.k.a. non-asymptotic convergence rate), in part due to the difficulty posed by the nonlinear metric. Only very recently Zhang and Sra [38] developed the first global complexity analysis of batch and stochastic gradient methods for geodesically convex functions. However, the batch and stochastic gradient methods in [38] suffer from problems similar to their vector space counterparts. For solving finite sum problems with $n$ components, the full-gradient method requires $n$ derivatives at each step; the stochastic method requires only one derivative but at the expense of slower $O(1/\epsilon^2)$ convergence to an $\epsilon$-accurate solution.

These issues have motivated much of the recent progress on faster stochastic optimization in vector spaces by using variance reduction [10; 18; 28] techniques. However, all ensuing methods critically rely on properties of vector spaces, whereby, adapting them to work in the context of Riemannian manifolds poses major challenges. Given the richness of Riemannian optimization (it includes vector space optimization as a special case) and its growing number of applications, developing fast stochastic Riemannian optimization is important. It will help us apply Riemannian optimization to large-scale problems, while offering a new set of algorithmic tools for the practitioner's repertoire.

**Contributions.**   We summarize the key contributions of this paper below.

- We introduce Riemannian SVRG (RSVRG), a variance reduced Riemannian stochastic gradient method based on SVRG [18]. We analyze RSVRG for geodesically strongly convex functions through a novel theoretical analysis that accounts for the nonlinear (curved) geometry of the manifold to yield linear convergence rates.
- Building on recent advances in variance reduction for nonconvex optimization [3; 25], we generalize the convergence analysis of RSVRG to (geodesically) nonconvex functions and also to gradient dominated functions (see §2 for the definition). Our analysis provides the first stochastic Riemannian method that is provably superior to both batch and stochastic (Riemannian) gradient methods for nonconvex finite-sum problems.
- Using a Riemannian formulation and applying our result for (geodesically) gradient-dominated functions, we provide new insights, and a short transparent analysis explaining fast convergence of variance reduced PCA for computing the leading eigenvector of a symmetric matrix.

To our knowledge, this paper provides the first stochastic gradient method with global linear convergence rates for geodesically strongly convex functions, as well as the first non-asymptotic convergence rates for geodesically nonconvex optimization (even in the batch case). Our analysis reveals how manifold geometry, in particular curvature, impacts convergence rates. We illustrate the benefits of RSVRG by showing an application to computing leading eigenvectors of a symmetric matrix and to the task of computing the Riemannian centroid of covariance matrices, a problem that has received great attention in the literature [5; 16; 38].

**Related Work.**   Variance reduction techniques, such as *control variates*, are widely used in Monte Carlo simulations [27]. In linear spaces, variance reduced methods for solving finite-sum problems have recently witnessed a huge surge of interest [e.g. 4; 10; 14; 18; 20; 28; 36]. They have been shown to accelerate stochastic optimization for strongly convex objectives, convex objectives, nonconvex $f_i$ ($i \in [n]$), and even when both $f$ and $f_i$ ($i \in [n]$) are nonconvex [3; 25]. Reddi et al. [25] further proved global linear convergence for gradient dominated nonconvex problems. Our analysis is inspired by [18; 25], but applies to the substantially more general Riemannian optimization setting.

References of Riemannian optimization can be found in [1; 33], where analysis is limited to asymptotic convergence (except [33, Theorem 4.2] which proved linear rate convergence for first-order line search method with bounded and positive definite hessian). Stochastic Riemannian optimization has

been previously considered in [7; 21], though with only asymptotic convergence analysis, and without any rates. Many applications of Riemannian optimization are known, including matrix factorization on fixed-rank manifold [32; 34], dictionary learning [8; 31], optimization under orthogonality constraints [11; 22], covariance estimation [35], learning elliptical distributions [30; 39], and Gaussian mixture models [15]. Notably, some nonconvex Euclidean problems are geodesically convex, for which Riemannian optimization can provide similar guarantees to convex optimization. Zhang and Sra [38] provide the first global complexity analysis for first-order Riemannian algorithms, but their analysis is restricted to geodesically convex problems with full or stochastic gradients. In contrast, we propose RSVRG, a variance reduced Riemannian stochastic gradient algorithm, and analyze its global complexity for both geodesically convex and nonconvex problems.

In parallel with our work, [19] also proposed and analyzed RSVRG specifically for the Grassmann manifold. Their complexity analysis is restricted to *local* convergence to strict local minima, which essentially corresponds to our analysis of (locally) geodesically strongly convex functions.

## 2   Preliminaries

Before formally discussing Riemannian optimization, let us recall some foundational concepts of Riemannian geometry. For a thorough review one can refer to any classic text, e.g.,[24].

A *Riemannian manifold* $(\mathcal{M}, \mathfrak{g})$ is a real smooth manifold $\mathcal{M}$ equipped with a Riemannain metric $\mathfrak{g}$. The metric $\mathfrak{g}$ induces an inner product structure in each tangent space $T_x\mathcal{M}$ associated with every $x \in \mathcal{M}$. We denote the inner product of $u, v \in T_x\mathcal{M}$ as $\langle u, v \rangle \triangleq \mathfrak{g}_x(u, v)$; and the norm of $u \in T_x\mathcal{M}$ is defined as $\|u\| \triangleq \sqrt{\mathfrak{g}_x(u, u)}$. The angle between $u, v$ is defined as $\arccos \frac{\langle u,v \rangle}{\|u\|\|v\|}$. A geodesic is a constant speed curve $\gamma : [0, 1] \to \mathcal{M}$ that is locally distance minimizing. An exponential map $\mathrm{Exp}_x : T_x\mathcal{M} \to \mathcal{M}$ maps $v$ in $T_x\mathcal{M}$ to $y$ on $\mathcal{M}$, such that there is a geodesic $\gamma$ with $\gamma(0) = x, \gamma(1) = y$ and $\dot{\gamma}(0) \triangleq \frac{d}{dt}\gamma(0) = v$. If between any two points in $\mathcal{X} \subset \mathcal{M}$ there is a unique geodesic, the exponential map has an inverse $\mathrm{Exp}_x^{-1} : \mathcal{X} \to T_x\mathcal{M}$ and the geodesic is the unique shortest path with $\|\mathrm{Exp}_x^{-1}(y)\| = \|\mathrm{Exp}_y^{-1}(x)\|$ the geodesic distance between $x, y \in \mathcal{X}$.

Parallel transport $\Gamma_x^y : T_x\mathcal{M} \to T_y\mathcal{M}$ maps a vector $v \in T_x\mathcal{M}$ to $\Gamma_x^y v \in T_y\mathcal{M}$, while preserving norm, and roughly speaking, "direction," analogous to translation in $\mathbb{R}^d$. A tangent vector of a geodesic $\gamma$ remains tangent if parallel transported along $\gamma$. Parallel transport preserves inner products.

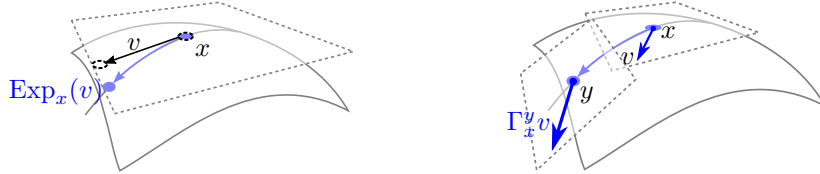

Figure 1: Illustration of manifold operations. (Left) A vector $v$ in $T_x\mathcal{M}$ is mapped to $\mathrm{Exp}_x(v)$; (right) A vector $v$ in $T_x\mathcal{M}$ is parallel transported to $T_y\mathcal{M}$ as $\Gamma_x^y v$.

The geometry of a Riemannian manifold is determined by its Riemannian metric tensor through various characterization of curvatures. Let $u, v \in T_x\mathcal{M}$ be linearly independent, so that they span a two dimensional subspace of $T_x\mathcal{M}$. Under the exponential map, this subspace is mapped to a two dimensional submanifold of $\mathcal{U} \subset \mathcal{M}$. The sectional curvature $\kappa(x, \mathcal{U})$ is defined as the Gauss curvature of $\mathcal{U}$ at $x$. As we will mainly analyze manifold trigonometry, for worst-case analysis, it is sufficient to consider sectional curvature.

**Function Classes.**   We now define some key terms. A set $\mathcal{X}$ is called *geodesically convex* if for any $x, y \in \mathcal{X}$, there is a geodesic $\gamma$ with $\gamma(0) = x, \gamma(1) = y$ and $\gamma(t) \in \mathcal{X}$ for $t \in [0, 1]$. Throughout the paper, we assume that the function $f$ in (1) is defined on a geodesically convex set $\mathcal{X}$ on a Riemannian manifold $\mathcal{M}$.

We call a function $f : \mathcal{X} \to \mathbb{R}$ *geodesically convex* (g-convex) if for any $x, y \in \mathcal{X}$ and any geodesic $\gamma$ such that $\gamma(0) = x, \gamma(1) = y$ and $\gamma(t) \in \mathcal{X}$ for $t \in [0, 1]$, it holds that

$$f(\gamma(t)) \leq (1 - t)f(x) + tf(y).$$

It can be shown that if the inverse exponential map is well-defined, an equivalent definition is that for any $x, y \in \mathcal{X}$, $f(y) \geq f(x) + \langle g_x, \mathrm{Exp}_x^{-1}(y) \rangle$, where $g_x$ is a subgradient of $f$ at $x$ (or the gradient if $f$ is differentiable). A function $f : \mathcal{X} \to \mathbb{R}$ is called *geodesically $\mu$-strongly convex* ($\mu$-strongly g-convex) if for any $x, y \in \mathcal{X}$ and subgradient $g_x$, it holds that

$$f(y) \geq f(x) + \langle g_x, \mathrm{Exp}_x^{-1}(y) \rangle + \tfrac{\mu}{2}\|\mathrm{Exp}_x^{-1}(y)\|^2.$$

We call a vector field $g : \mathcal{X} \to \mathbb{R}^d$ *geodesically $L$-Lipschitz* ($L$-g-Lipschitz) if for any $x, y \in \mathcal{X}$,

$$\|g(x) - \Gamma_y^x g(y)\| \leq L\|\mathrm{Exp}_x^{-1}(y)\|,$$

where $\Gamma_y^x$ is the parallel transport from $y$ to $x$. We call a differentiable function $f : \mathcal{X} \to \mathbb{R}$ *geodesically $L$-smooth* ($L$-g-smooth) if its gradient is $L$-g-Lipschitz, in which case we have

$$f(y) \leq f(x) + \langle g_x, \mathrm{Exp}_x^{-1}(y) \rangle + \tfrac{L}{2}\|\mathrm{Exp}_x^{-1}(y)\|^2.$$

We say $f : \mathcal{X} \to \mathbb{R}$ is *$\tau$-gradient dominated* if $x^*$ is a global minimizer of $f$ and for every $x \in \mathcal{X}$

$$f(x) - f(x^*) \leq \tau \|\nabla f(x)\|^2. \tag{2}$$

We recall the following trigonometric distance bound that is essential for our analysis:

**Lemma 1** ([7; 38]). If $a, b, c$ are the side lengths of a geodesic triangle in a Riemannian manifold with sectional curvature lower bounded by $\kappa_{\min}$, and $A$ is the angle between sides $b$ and $c$ (defined through inverse exponential map and inner product in tangent space), then

$$a^2 \leq \frac{\sqrt{|\kappa_{\min}|}c}{\tanh(\sqrt{|\kappa_{\min}|}c)} b^2 + c^2 - 2bc\cos(A). \tag{3}$$

An *Incremental First-order Oracle (IFO)* [2] in (1) takes an $i \in [n]$ and a point $x \in \mathcal{X}$, and returns a pair $(f_i(x), \nabla f_i(x)) \in \mathbb{R} \times T_x \mathcal{M}$. We measure non-asymptotic complexity in terms of IFO calls.

# 3 Riemannian SVRG

In this section we introduce RSVRG formally. We make the following standing assumptions: (a) $f$ attains its optimum at $x^* \in \mathcal{X}$; (b) $\mathcal{X}$ is compact, and the diameter of $\mathcal{X}$ is bounded by $D$, that is, $\max_{x,y \in \mathcal{X}} d(x,y) \leq D$; (c) the sectional curvature in $\mathcal{X}$ is *upper* bounded by $\kappa_{\max}$, and within $\mathcal{X}$ the exponential map is invertible; and (d) the sectional curvature in $\mathcal{X}$ is *lower* bounded by $\kappa_{\min}$. We define the following key geometric constant that capture the impact of manifold curvature:

$$\zeta = \begin{cases} \frac{\sqrt{|\kappa_{\min}|}D}{\tanh(\sqrt{|\kappa_{\min}|}D)}, & \text{if } \kappa_{\min} < 0, \\ 1, & \text{if } \kappa_{\min} \geq 0, \end{cases} \tag{4}$$

We note that most (if not all) practical manifold optimization problems can satisfy these assumptions.

Our proposed RSVRG algorithm is shown in Algorithm 1. Compared with the Euclidean SVRG, it differs in two key aspects: the variance reduction step uses parallel transport to combine gradients from different tangent spaces; and the exponential map is used (instead of the update $x_t^{s+1} - \eta v_t^{s+1}$).

## 3.1 Convergence analysis for strongly g-convex functions

In this section, we analyze global complexity of RSVRG for solving (1), where each $f_i$ ($i \in [n]$) is g-smooth and $f$ is strongly g-convex. In this case, we show that RSVRG has linear convergence rate. This is in contrast with the $O(1/t)$ rate of Riemannian stochastic gradient algorithm for strongly g-convex functions [38].

**Theorem 1.** Assume in (1) each $f_i$ is $L$-g-smooth, and $f$ is $\mu$-strongly g-convex, then if we run Algorithm 1 with Option I and parameters that satisfy

$$\alpha = \frac{3\zeta\eta L^2}{\mu - 2\zeta\eta L^2} + \frac{(1 + 4\zeta\eta^2 - 2\eta\mu)^m(\mu - 5\zeta\eta L^2)}{\mu - 2\zeta\eta L^2} < 1$$

then with $S$ outer loops, the Riemannian SVRG algorithm produces an iterate $x_a$ that satisfies

$$\mathbb{E}d^2(x_a, x^*) \leq \alpha^S d^2(x^0, x^*).$$

---

**Algorithm 1:** RSVRG $(x^0, m, \eta, S)$

---

**Parameters:** update frequency $m$, learning rate $\eta$, number of epochs $S$
initialize $\tilde{x}^0 = x^0$;
**for** $s = 0, 1, \ldots, S - 1$ **do**
      $x_0^{s+1} = \tilde{x}^s$;
      $g^{s+1} = \frac{1}{n} \sum_{i=1}^n \nabla f_i(\tilde{x}^s)$;
      **for** $t = 0, 1, \ldots, m - 1$ **do**
          Randomly pick $i_t \in \{1, \ldots, n\}$;
          $v_t^{s+1} = \nabla f_{i_t}(x_t^{s+1}) - \Gamma_{\tilde{x}^s}^{x_t^{s+1}} \left( \nabla f_{i_t}(\tilde{x}^s) - g^{s+1} \right)$;
          $x_{t+1}^{s+1} = \mathrm{Exp}_{x_t^{s+1}} \left( -\eta v_t^{s+1} \right)$;
      **end**
      Set $\tilde{x}^{s+1} = x_m^{s+1}$;
**end**

**Option I:** output $x_a = \tilde{x}^S$;
**Option II:** output $x_a$ chosen uniformly randomly from $\{\{x_t^{s+1}\}_{t=0}^{m-1}\}_{s=0}^{S-1}$.

---

The proof of Theorem 1 is in the appendix, and takes a different route compared with the original SVRG proof [18]. Specifically, due to the nonlinear Riemannian metric, we are not able to bound the squared norm of the variance reduced gradient by $f(x) - f(x^*)$. Instead, we bound this quantity by the squared distances to the minimizer, and show linear convergence of the iterates. A bound on $\mathbb{E}[f(x) - f(x^*)]$ is then implied by $L$-g-smoothness, albeit with a stronger dependence on the condition number. Theorem 1 leads to the following more digestible corollary on the global complexity of the algorithm:

**Corollary 1.** With assumptions as in Theorem 1 and properly chosen parameters, after $O\left( (n + \frac{\zeta L^2}{\mu^2}) \log(\frac{1}{\epsilon}) \right)$ IFO calls, the output $x_a$ satisfies

$$\mathbb{E}[f(x_a) - f(x^*)] \le \epsilon.$$

We give a proof with specific parameter choices in the appendix. Observe the dependence on $\zeta$ in our result: for $\kappa_{\min} < 0$, we have $\zeta > 1$, which implies that negative space curvature adversarially affects convergence rate; while for $\kappa_{\min} \ge 0$, we have $\zeta = 1$, which implies that for nonnegatively curved manifolds, the impact of curvature is not explicit. In the rest of our analysis we will see a similar effect of sectional curvature; this phenomenon seems innate to manifold optimization (also see [38]).

In the analysis we do not assume each $f_i$ to be g-convex, which resulted in a worse dependence on the condition number. We note that a similar result was obtained in linear space [12]. However, we will see in the next section that by generalizing the analysis for gradient dominated functions in [25], we are able to greatly improve this dependence.

### 3.2 Convergence analysis for geodesically nonconvex functions

In this section, we analyze global complexity of RSVRG for solving (1), where each $f_i$ is only required to be $L$-g-smooth, and neither $f_i$ nor $f$ need be g-convex. We measure convergence to a stationary point using $\|\nabla f(x)\|^2$ following [13]. Note, however, that here $\nabla f(x) \in T_x \mathcal{M}$ and $\|\nabla f(x)\|$ is defined via the inner product in $T_x \mathcal{M}$. We first note that Riemannian-SGD on nonconvex $L$-g-smooth problems attains $O(1/\epsilon^2)$ convergence as SGD [13] holds; we relegate the details to the appendix.

Recently, two groups independently proved that variance reduction also benefits stochastic gradient methods for nonconvex smooth finite-sum optimization problems, with different analysis [3; 25]. Our analysis for nonconvex RSVRG is inspired by [25]. Our main result for this section is Theorem 2.

**Theorem 2.** Assume in (1) each $f_i$ is $L$-g-smooth, the sectional curvature in $\mathcal{X}$ is lower bounded by $\kappa_{\min}$, and we run Algorithm 1 with Option II. Then there exist universal constants $\mu_0 \in (0, 1), \nu > 0$ such that if we set $\eta = \mu_0/(L n^{\alpha_1} \zeta^{\alpha_2})$ ($0 < \alpha_1 \le 1$ and $0 \le \alpha_2 \le 2$), $m = \lfloor n^{3\alpha_1/2}/(3\mu_0 \zeta^{1-2\alpha_2}) \rfloor$ and $T = mS$, we have

$$\mathbb{E}[\|\nabla f(x_a)\|^2] \le \frac{L n^{\alpha_1} \zeta^{\alpha_2}[f(x^0) - f(x^*)]}{T\nu},$$

where $x^*$ is an optimal solution to (1).

**Algorithm 2:** GD-SVRG($x^0, m, \eta, S, K$)

---
**Parameters:** update frequency $m$, learning rate $\eta$, number of epochs $S$, $K$, $x^0$
**for** $k = 0, \ldots, K-1$ **do**
 | $x^{k+1} = \text{RSVRG}(x^k, m, \eta, S)$ with Option II;
**end**
**Output:** $x^K$

---

The key challenge in proving Theorem 2 in the Riemannian setting is to incorporate the impact of using a nonlinear metric. Similar to the g-convex case, the nonlinear metric impacts the convergence, notably through the constant $\zeta$ that depends on a lower-bound on sectional curvature.

Reddi et al. [25] suggested setting $\alpha_1 = 2/3$, in which case we obtain the following corollary.

**Corollary 2.** With assumptions and parameters in Theorem 2, choosing $\alpha_1 = 2/3$, the IFO complexity for achieving an $\epsilon$-accurate solution is:

$$\text{IFO calls} = \begin{cases} O\left(n + (n^{2/3}\zeta^{1-\alpha_2}/\epsilon)\right), & \text{if } \alpha_2 \leq 1/2, \\ O\left(n\zeta^{2\alpha_2-1} + (n^{2/3}\zeta^{\alpha_2}/\epsilon)\right), & \text{if } \alpha_2 > 1/2. \end{cases}$$

Setting $\alpha_2 = 1/2$ in Corollary 2 immediately leads to Corollary 3:

**Corollary 3.** With assumptions in Theorem 2 and $\alpha_1 = 2/3, \alpha_2 = 1/2$, the IFO complexity for achieving an $\epsilon$-accurate solution is $O\left(n + (n^{2/3}\zeta^{1/2}/\epsilon)\right)$.

The same reasoning allows us to also capture the class of gradient dominated functions (2), for which Reddi et al. [25] proved that SVRG converges linearly to a global optimum. We have the following corresponding theorem for RSVRG:

**Theorem 3.** Suppose that in addition to the assumptions in Theorem 2, $f$ is $\tau$-gradient dominated. Then there exist universal constants $\mu_0 \in (0,1), \nu > 0$ such that if we run Algorithm 2 with $\eta = \mu_0/(Ln^{2/3}\zeta^{1/2}), m = \lfloor n/(3\mu_0)\rfloor, S = \lceil(6 + \frac{18\mu_0}{n-3})L\tau\zeta^{1/2}\mu_0/(\nu n^{1/3})\rceil$, we have

$$\mathbb{E}[\|\nabla f(x^K)\|^2] \leq 2^{-K}\|\nabla f(x^0)\|^2,$$
$$\mathbb{E}[f(x^K) - f(x^*)] \leq 2^{-K}[f(x^0) - f(x^*)].$$

We summarize the implication of Theorem 3 as follows (note the dependence on curvature):

**Corollary 4.** With Algorithm 2 and the parameters in Theorem 3, the IFO complexity to compute an $\epsilon$-accurate solution for a gradient dominated function $f$ is $O((n + L\tau\zeta^{1/2}n^{2/3})\log(1/\epsilon))$.

A typical example of gradient dominated function is a strongly g-convex function (see appendix). Specifically, we have the following corollary, which prove linear convergence rate of RSVRG with the same assumptions as in Theorem 1, improving the dependence on the condition number.

**Corollary 5.** With Algorithm 2 and the parameters in Theorem 3, the IFO complexity to compute an $\epsilon$-accurate solution for a $\mu$-strongly g-convex function $f$ is $O((n + \mu^{-1}L\zeta^{1/2}n^{2/3})\log(1/\epsilon))$.

## 4 Applications

### 4.1 Computing the leading eigenvector

In this section, we apply our analysis of RSVRG for gradient dominated functions (Theorem 3) to fast eigenvector computation, a fundamental problem that is still being actively researched in the big-data setting [12; 17; 29]. For the problem of computing the leading eigenvector, i.e.,

$$\min_{x^\top x = 1} \quad -x^\top\left(\sum_{i=1}^n z_i z_i^\top\right)x \quad \triangleq \quad -x^\top A x = f(x), \tag{5}$$

existing analyses for state-of-the-art algorithms typically result in $O(1/\delta^2)$ dependence on the eigengap $\delta$ of $A$, as opposed to the conjectured $O(1/\delta)$ dependence [29], as well as the $O(1/\delta)$ dependence of power iteration. Here we give new support for the $O(1/\delta)$ conjecture. Note that Problem (5) seen as one in $\mathbb{R}^d$ is nonconvex, with negative semidefinite Hessian everywhere, and has nonlinear constraints. However, we show that on the hypersphere $\mathbb{S}^{d-1}$ Problem (5) is unconstrained, and has *gradient dominated* objective. In particular we have the following result:

**Theorem 4.** Suppose $A$ has eigenvalues $\lambda_1 > \lambda_2 \geq \cdots \geq \lambda_d$ and $\delta = \lambda_1 - \lambda_2$, and $x^0$ is drawn uniformly randomly on the hypersphere. Then with probability $1 - p$, $x^0$ falls in a Riemannian ball of a global optimum of the objective function, within which the objective function is $O(\frac{d}{p^2\delta})$-gradient dominated.

We provide the proof of Theorem 4 in appendix. Theorem 4 gives new insights for why the conjecture might be true – once it is shown that with a constant stepsize and with high probability (both independent of $\delta$) the iterates remain in such a Riemannian ball, applying Corollary 4 one can immediately prove the $O(1/\delta)$ dependence conjecture. We leave this analysis as future work.

Next we show that variance reduced PCA (VR-PCA) [29] is closely related to RSVRG. We implement Riemannian SVRG for PCA, and use the code for VR-PCA in [29]. Analytic forms for exponential map and parallel transport on hypersphere can be found in [1, Example 5.4.1; Example 8.1.1]. We conduct well-controlled experiments comparing the performance of two algorithms. Specifically, to investigate the dependence of convergence on $\delta$, for each $\delta = 10^{-3}/k$ where $k = 1, \ldots, 25$, we generate a $d \times n$ matrix $Z = (z_1, \ldots, z_n)$ where $d = 10^3, n = 10^4$ using the method $Z = UDV^\top$ where $U, V$ are orthonormal matrices and $D$ is a diagonal matrix, as described in [29]. Note that $A$ has the same eigenvalues as $D^2$. All the data matrices share the same $U, V$ and only differ in $\delta$ (thus also in $D$). We also fix the same random initialization $x^0$ and random seed. We run both algorithms on each matrix for 50 epochs. For every five epochs, we estimate the number of epochs required to double its accuracy [2]. This number can serve as an indicator of the global complexity of the algorithm. We plot this number for different epochs against $1/\delta$, shown in Figure 2. Note that the performance of RSVRG and VR-PCA with the same stepsize is very similar, which implies a close connection of the two. Indeed, the update $\frac{x+v}{\|x+v\|}$ used in [29] and others is a well-known approximation to the exponential map $\mathrm{Exp}_x(v)$ with small stepsize (a.k.a. retraction). Also note the complexity of both algorithms seems to have an asymptotically linear dependence on $1/\delta$.

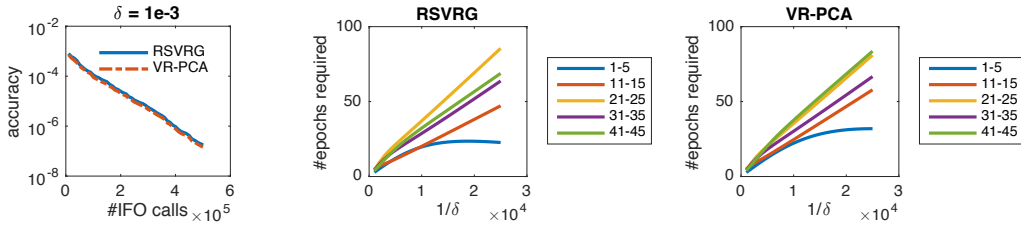

Figure 2: **Computing the leading eigenvector.** Left: RSVRG and VR-PCA are indistinguishable in terms of IFO complexity. Middle and right: Complexity appears to depend on $1/\delta$. $x$-axis shows the inverse of eigengap $\delta$, $y$-axis shows the estimated number of epochs required to double the accuracy. Lines represent different epoch index. All variables are controlled except for $\delta$.

## 4.2 Computing the Riemannian centroid

In this subsection we validate that RSVRG converges linearly for averaging PSD matrices under the Riemannian metric. The problem for finding the Riemannian centroid of a set of PSD matrices $\{A_i\}_{i=1}^n$ is $X^* = \arg\min_{X \succeq 0} \left\{ f(X; \{A_i\}_{i=1}^n) \triangleq \sum_{i=1}^n \|\log(X^{-1/2}A_iX^{-1/2})\|_F^2 \right\}$ where $X$ is also a PSD matrix. This is a geodesically strongly convex problem, yet nonconvex in Euclidean space. It has been studied both in matrix computation and in various applications [5; 16]. We use the same experiment setting as described in [38] [3], and compare RSVRG against Riemannian full gradient (RGD) and stochastic gradient (RSGD) algorithms (Figure 3). Other methods for this problem include the relaxed Richardson iteration algorithm [6], the approximated joint diagonalization algorithm [9], and Riemannian Newton and quasi-Newton type methods, notably the limited-memory Riemannian

BFGS [37]. However, none of these methods were shown to greatly outperform RGD, especially in data science applications where $n$ is large and extremely small optimization error is not required.

Note that the objective is sum of squared Riemannian distances in a nonpositively curved space, thus is $(2n)$-strongly g-convex and $(2n\zeta)$-g-smooth. According to the proof of Corollary 1 (see appendix) the optimal stepsize for RSVRG is $O(1/(\zeta^3 n))$. For all the experiments, we initialize all the algorithms using the arithmetic mean of the matrices. We set $\eta = \frac{1}{100n}$, and choose $m = n$ in Algorithm 1 for RSVRG, and use suggested parameters from [38] for other algorithms. The results suggest RSVRG has clear advantage in the large scale setting.

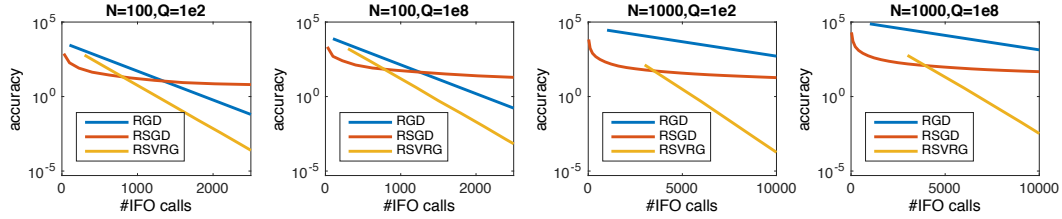

Figure 3: **Riemannian mean of PSD matrices.** $N$: number of matrices, $Q$: conditional number of each matrix. $x$-axis shows the actual number of IFO calls, $y$-axis show $f(X) - f(X^*)$ in log scale. Lines show the performance of different algorithms in colors. Note that RSVRG achieves linear convergence and is especially advantageous for large dataset.

## 5 Discussion

We introduce Riemannian SVRG, the first variance reduced stochastic gradient algorithm for Riemannian optimization. In addition, we analyze its global complexity for optimizing geodesically strongly convex, convex, and nonconvex functions, explicitly showing their dependence on sectional curvature. Our experiments validate our analysis that Riemannian SVRG is much faster than full gradient and stochastic gradient methods for solving finite-sum optimization problems on Riemannian manifolds.

Our analysis of computing the leading eigenvector as a Riemannian optimization problem is also worth noting: a nonconvex problem with nonpositive Hessian and nonlinear constraints in the ambient space turns out to be gradient dominated on the manifold. We believe this shows the promise of theoretical study of Riemannian optimization, and geometric optimization in general, and we hope it encourages other researchers in the community to join this endeavor.

Our work also has limitations – most practical Riemannian optimization algorithms use retraction and vector transport to efficiently approximate the exponential map and parallel transport, which we do not analyze in this work. A systematic study of retraction and vector transport is an important topic for future research. For other applications of Riemannian optimization such as low-rank matrix completion [34], covariance matrix estimation [35] and subspace tracking [11], we believe it would also be promising to apply fast incremental gradient algorithms in the large scale setting.

**Acknowledgment:** SS acknowledges support of NSF grant: IIS-1409802. HZ acknowledges support from the Leventhal Fellowship.

## Footnotes

[1]Riemannian optimization is optimization on a *known* manifold structure. Note the distinction from *manifold learning*, which attempts to learn a manifold structure from data. We briefly review some Riemannian optimization applications in the related work.

[2] Accuracy is measured by $\frac{f(x) - f(x^*)}{|f(x^*)|}$, i.e. the relative error between the objective value and the optimum. We measure how much the error has been reduced after each five epochs, which is a multiplicative factor $c < 1$ on the error at the start of each five epochs. Then we use $\log(2)/\log(1/c) * 5$ as the estimate, assuming $c$ stays constant.

[3] We generate $100 \times 100$ random PSD matrices using the Matrix Mean Toolbox [6] with normalization so that the norm of each matrix equals 1.

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
