[Supplementary Material · rsvrg_appendix_final.pdf]

# Appendix: Fast Stochastic Optimization on Riemannian Manifolds

## A  Proofs for Section 3.1

**Theorem 1.** Assume in (1) each $f_i$ is $L$-g-smooth, and $f$ is $\mu$-strongly g-convex, then if we run Algorithm 1 with Option I and parameters that satisfy

$$\alpha = \frac{3\zeta\eta L^2}{\mu - 2\zeta\eta L^2} + \frac{(1 + 4\zeta\eta^2 - 2\eta\mu)^m(\mu - 5\zeta\eta L^2)}{\mu - 2\zeta\eta L^2} < 1$$

then with $S$ outer loops, the Riemannian SVRG algorithm produces an iterate $x_a$ that satisfies

$$\mathbb{E}d^2(x_a, x^*) \le \alpha^S d^2(x^0, x^*).$$

*Proof.* We start by bounding the squared norm of the variance reduced gradient. Since $v_t^{s+1} = \nabla f_{i_t}(x_t^{s+1}) - \Gamma_{\tilde{x}^s}^{x_t^{s+1}}\left(\nabla f_{i_t}(\tilde{x}^s) - g^{s+1}\right)$, conditioned on $x_t^{s+1}$ and taking expectation with respect to $i_t$, we obtain:

$$
\begin{aligned}
\mathbb{E}\|v_t^{s+1}\|^2 &= \mathbb{E}\left\|\nabla f_{i_t}(x_t^{s+1}) - \Gamma_{\tilde{x}^s}^{x_t^{s+1}}\left(\nabla f_{i_t}(\tilde{x}^s) - g^{s+1}\right)\right\|^2 \\
&= \mathbb{E}\left\|\left(\nabla f_{i_t}(x_t^{s+1}) - \Gamma_{\tilde{x}^s}^{x_t^{s+1}}\nabla f_{i_t}(\tilde{x}^s)\right) + \Gamma_{\tilde{x}^s}^{x_t^{s+1}}\left(\nabla f(\tilde{x}^s) - \Gamma_{x^*}^{\tilde{x}^s}\nabla f(x^*)\right)\right\|^2 \\
&\le 2\mathbb{E}\left\|\nabla f_{i_t}(x_t^{s+1}) - \Gamma_{\tilde{x}^s}^{x_t^{s+1}}\nabla f_{i_t}(\tilde{x}^s)\right\|^2 + 2\mathbb{E}\left\|\Gamma_{\tilde{x}^s}^{x_t^{s+1}}\left(\nabla f(\tilde{x}^s) - \Gamma_{x^*}^{\tilde{x}^s}\nabla f(x^*)\right)\right\|^2 \\
&= 2\mathbb{E}\left\|\nabla f_{i_t}(x_t^{s+1}) - \Gamma_{\tilde{x}^s}^{x_t^{s+1}}\nabla f_{i_t}(\tilde{x}^s)\right\|^2 + 2\mathbb{E}\left\|\nabla f(\tilde{x}^s) - \Gamma_{x^*}^{\tilde{x}^s}\nabla f(x^*)\right\|^2 \\
&\le 2L^2\left\|\text{Exp}_{x_t^{s+1}}^{-1}(\tilde{x}^s)\right\|^2 + 2L^2\left\|\text{Exp}_{\tilde{x}^s}^{-1}(x^*)\right\|^2 \\
&\le 2L^2\left(\left\|\text{Exp}_{x_t^{s+1}}^{-1}(x^*)\right\| + \left\|\text{Exp}_{\tilde{x}^s}^{-1}(x^*)\right\|\right)^2 + 2L^2\left\|\text{Exp}_{\tilde{x}^s}^{-1}(x^*)\right\|^2 \\
&\le 4L^2\left\|\text{Exp}_{x_t^{s+1}}^{-1}(x^*)\right\|^2 + 6L^2\left\|\text{Exp}_{\tilde{x}^s}^{-1}(x^*)\right\|^2
\end{aligned}
$$

We use $\|a + b\|^2 \le 2\|a\|^2 + 2\|b\|^2$ twice, in the first and fourth inequalities. The second equality is due to $\nabla f(x^*) = 0$. The second inequality is due to the $L$-g-smoothness assumption. The third inequality is due to triangle inequality.

Notice that $\mathbb{E}v_t^{s+1} = \nabla f(x_t^{s+1})$ and $x_{t+1}^{s+1} = \text{Exp}_{x_t^{s+1}}(-\eta v_t^{s+1})$, we thus have

$$
\begin{aligned}
\mathbb{E}d^2(x_{t+1}^{s+1}, x^*) &\le d^2(x_t^{s+1}, x^*) + 2\eta\langle\text{Exp}_{x_t^{s+1}}^{-1}(x^*), \mathbb{E}v_t\rangle + \zeta\eta^2\mathbb{E}\|v_t\|^2 \\
&\le d^2(x_t^{s+1}, x^*) + 2\eta\langle\text{Exp}_{x_t^{s+1}}^{-1}(x^*), \nabla f(x_t^{s+1})\rangle \\
&\quad + \zeta\eta^2 L^2\left(4d^2(x_t^{s+1}, x^*) + 6d^2(\tilde{x}^s, x^*)\right) \\
&\le \left(1 + 4\zeta\eta^2 L^2 - \eta\mu\right)d^2(x_t^{s+1}, x^*) + 6\zeta\eta^2 L^2 d^2(\tilde{x}^s, x^*) \\
&\quad + 2\eta\left(f(x^*) - f(x_t^{s+1})\right) \\
&\le \left(1 + 4\zeta\eta^2 L^2 - 2\eta\mu\right)d^2(x_t^{s+1}, x^*) + 6\zeta\eta^2 L^2 d^2(\tilde{x}^s, x^*)
\end{aligned}
$$

The first inequality uses the trigonometric distance lemma, the second one uses previously obtained bound for $\mathbb{E}\|v_t\|^2$, the third and fourth use the $\mu$-strong g-convexity of $f(x)$.

We now denote $u_t \triangleq \mathbb{E}d^2(x_t^{s+1}, x^*), q \triangleq 1 + 4\zeta\eta^2 L^2 - 2\eta\mu, p \triangleq 6\zeta\eta^2 L^2/(1 - q)$. Hence by taking expectation with all the history, and noting $\tilde{x}^s = x_0^{s+1}$, we have $u_{t+1} \le qu_t + p(1 - q)u_0$, i.e. $u_{t+1} - pu_0 \le q(u_t - pu_0)$. Therefore, $u_m - pu_0 \le q^m(u_0 - pu_0)$, hence we get

$$u_m \le (p + q^m(1 - p))u_0,$$

where $p + q^m(1 - p) = \frac{3\zeta\eta L^2}{\mu - 2\zeta\eta L^2} + \frac{(1 + 4\zeta\eta^2 L^2 - 2\eta\mu)^m(\mu - 5\zeta\eta L^2)}{\mu - 2\zeta\eta L^2} = \alpha$. It follows directly from the algorithm that after $S$ outer loops, $\mathbb{E}d^2(x_a, x^*) = \mathbb{E}d^2(\tilde{x}^S, x^*) \le \alpha^S d^2(x^0, x^*)$.  $\square$

**Corollary 1.** With assumptions as in Theorem 1 and properly chosen parameters, after $O\left(\left(n + \frac{\zeta L^2}{\mu^2}\right)\log(\frac{1}{\epsilon})\right)$ IFO calls, the output $x_a$ satisfies

$$\mathbb{E}[f(x_a) - f(x^*)] \leq \epsilon.$$

*Proof.* Assume we choose $\eta = \mu/(17\zeta L^2)$ and $m \geq 10\zeta L^2/\mu^2$, it follows that $q = 1 - 30\mu^2/(289\zeta L^2) \leq 1 - \mu^2/(10\zeta L^2), p = 1/5$ and therefore

$$u_m \leq \left(\frac{1}{5} + \frac{4}{5}\left(1 - \mu^2/(10\zeta L^2)\right)^{10\zeta L^2/\mu^2}\right) u_0 \leq \left(\frac{1}{5} + \frac{4}{5e}\right) u_0 \leq \frac{u_0}{2},$$

where the second inequality is due to $(1-x)^{1/x} \leq 1/e$ for $x \in (0,1)$. Applying Theorem 1 with $\alpha = 1/2$, we have $\mathbb{E}d^2(x_a, x^*) \leq 2^{-S}d^2(x^0, x^*)$. Note that by using the $L$-g-smooth assumption, we also get $\mathbb{E}[f(x_a) - f(x^*)] \leq \mathbb{E}\left[\frac{1}{2}Ld^2(x_a, x^*)\right] \leq 2^{-S-1}Ld^2(x^0, x^*)$. It thus suffices to run $\log_2(Ld^2(x^0, x^*)/\epsilon) - 1$ outer loops to guarantee $\mathbb{E}[f(x_a) - f(x^*)] \leq \epsilon$.

For the $s$-th outer loop, we need $n$ IFO calls to evaluate the full gradient at $\tilde{x}^s$, and $2m$ IFO calls when calculating each variance reduced gradient. Hence the total number of IFO calls to reach $\epsilon$ accuracy is $O\left(\left(n + \frac{\zeta L^2}{\mu^2}\right)\log(\frac{1}{\epsilon})\right)$. $\qquad\square$

## B  Proofs for Section 3.2

**Theorem 5.** Assuming the inverse exponential map is well-defined on $\mathcal{X}$, $f : \mathcal{X} \to \mathbb{R}$ is a geodesically $L$-smooth function, stochastic first-order oracle $\nabla \tilde{f}(x)$ satisfies $\mathbb{E}[\nabla \tilde{f}(x^t)] = \nabla f(x^t), \|\nabla \tilde{f}(x^t)\|^2 \leq \sigma^2$, then the SGD algorithm $x^{t+1} = \mathrm{Exp}_{x^t}(-\eta\nabla \tilde{f}(x^t))$ with $\eta = c/\sqrt{T}, c = \sqrt{\frac{2(f(x^0)-f(x^*))}{L\sigma^2}}$ satisfies

$$\min_{0 \leq t \leq T-1} \mathbb{E}[\|\nabla f(x^t)\|^2] \leq \sqrt{\frac{2(f(x^0)-f(x^*))L}{T}}\sigma.$$

*Proof.*

$$\mathbb{E}[f(x^{t+1})] \leq \mathbb{E}[f(x^t) + \langle\nabla f(x^t), \mathrm{Exp}_{x^t}^{-1}(x^{t+1})\rangle + \frac{L}{2}\|\mathrm{Exp}_{x^t}^{-1}(x^{t+1})\|^2]$$

$$\leq \mathbb{E}[f(x^t)] - \eta\mathbb{E}[\|\nabla f(x^t)\|^2] + \frac{L\eta^2}{2}\mathbb{E}[\|\nabla \tilde{f}(x^t)\|^2]$$

$$\leq \mathbb{E}[f(x^t)] - \eta\mathbb{E}[\|\nabla f(x^t)\|^2] + \frac{L\eta^2}{2}\sigma^2$$

After rearrangement, we obtain

$$\mathbb{E}[\|\nabla f(x^t)\|^2] \leq \frac{1}{\eta}\mathbb{E}[f(x^t) - f(x^{t+1})] + \frac{L\eta}{2}\sigma^2$$

Summing up the above equation from $t = 0$ to $T - 1$ and using $\eta = c/\sqrt{T}$ where

$$c = \sqrt{\frac{2(f(x^0)-f(x^*))}{L\sigma^2}}$$

we obtain

$$\min_t \mathbb{E}[\|\nabla f(x^t)\|^2] \leq \frac{1}{T}\sum_{t=0}^{T-1}\mathbb{E}[\|f(x^t)\|^2] \leq \frac{1}{T\eta}\mathbb{E}[f(x^0) - f(x^T)] + \frac{L\eta}{2}\sigma^2$$

$$\leq \frac{1}{T\eta}(f(x^0) - f(x^*)) + \frac{L\eta}{2}\sigma^2$$

$$\leq \sqrt{\frac{2(f(x^0)-f(x^*))L}{T}}\sigma$$

$\qquad\square$

**Lemma 2.** Assume in (1) each $f_i$ is $L$-g-smooth, the sectional curvature in $\mathcal{X}$ is lower bounded by $\kappa_{\min}$, and we run Algorithm 1 with Option II. For $c_t, c_{t+1}, \beta, \eta > 0$, suppose we have

$$c_t = c_{t+1}\left(1 + \beta\eta + 2\zeta L^2\eta^2\right) + L^3\eta^2$$

and

$$\delta(t) = \eta - \frac{c_{t+1}\eta}{\beta} - L\eta^2 - 2c_{t+1}\zeta\eta^2 > 0,$$

then the iterate $x_t^{s+1}$ satisfies the bound:

$$\mathbb{E}\left[\|\nabla f(x_t^{s+1})\|^2\right] \leq \frac{R_t^{s+1} - R_{t+1}^{s+1}}{\delta_t}$$

where $R_t^{s+1} := \mathbb{E}[f(x_t^{s+1}) + c_t\|\mathrm{Exp}_{\tilde{x}^s}(x_t^{s+1})\|^2]$ for $0 \leq s \leq S - 1$.

*Proof.* Since $f$ is $L$-smooth we have

$$\mathbb{E}[f(x_{t+1}^{s+1})] \leq \mathbb{E}[f(x_t^{s+1}) + \langle \nabla f(x_t^{s+1}), \mathrm{Exp}_{x_t^{s+1}}^{-1}(x_{t+1}^{s+1})\rangle + \frac{L}{2}\|\mathrm{Exp}_{x_t^{s+1}}^{-1}(x_{t+1}^{s+1})\|^2]$$

$$\leq \mathbb{E}[f(x_t^{s+1}) - \eta\|\nabla f(x_t^{s+1})\|^2 + \frac{L\eta^2}{2}\|v_t^{s+1}\|^2] \tag{6}$$

Consider now the Lyapunov function

$$R_t^{s+1} := \mathbb{E}[f(x_t^{s+1}) + c_t\|\mathrm{Exp}_{\tilde{x}^s}(x_t^{s+1})\|^2]$$

For bounding it we will require the following:

$$\mathbb{E}[\|\mathrm{Exp}_{\tilde{x}^s}^{-1}(x_{t+1}^{s+1})\|^2] \leq \mathbb{E}[\|\mathrm{Exp}_{\tilde{x}^s}^{-1}(x_t^{s+1})\|^2 + \zeta\|\mathrm{Exp}_{x_t^{s+1}}^{-1}(x_{t+1}^{s+1})\|^2$$

$$- 2\langle \mathrm{Exp}_{x_t^{s+1}}^{-1}(x_{t+1}^{s+1}), \mathrm{Exp}_{x_t^{s+1}}^{-1}(\tilde{x}^s)\rangle]$$

$$= \mathbb{E}[\|\mathrm{Exp}_{\tilde{x}^s}^{-1}(x_t^{s+1})\|^2 + \zeta\eta^2\|v_t^{s+1}\|^2$$

$$+ 2\eta\langle \nabla f(x_t^{s+1}), \mathrm{Exp}_{x_t^{s+1}}^{-1}(\tilde{x}^s)\rangle]$$

$$\leq \mathbb{E}[\|\mathrm{Exp}_{\tilde{x}^s}^{-1}(x_t^{s+1})\|^2 + \zeta\eta^2\|v_t^{s+1}\|^2]$$

$$+ 2\eta\mathbb{E}\left[\frac{1}{2\beta}\|\nabla f(x_t^{s+1})\|^2 + \frac{\beta}{2}\|\mathrm{Exp}_{\tilde{x}^s}^{-1}(x_t^{s+1})\|^2\right] \tag{7}$$

where the first inequality is due to Lemma 1, the second due to $2\langle a, b\rangle \leq \frac{1}{\beta}\|a\|^2 + \beta\|b\|^2$. Plugging Equation (6) and Equation (7) into $R_{t+1}^{s+1}$, we obtain the following bound:

$$R_{t+1}^{s+1} \leq \mathbb{E}[f(x_t^{s+1}) - \eta\|\nabla f(x_t^{s+1})\|^2 + \frac{L\eta^2}{2}\|v_t^{s+1}\|^2]$$

$$+ c_{t+1}\mathbb{E}[\|\mathrm{Exp}_{\tilde{x}^s}^{-1}(x_t^{s+1})\|^2 + \zeta\eta^2\|v_t^{s+1}\|^2]$$

$$+ 2c_{t+1}\eta\mathbb{E}\left[\frac{1}{2\beta}\|\nabla f(x_t^{s+1})\|^2 + \frac{\beta}{2}\|\mathrm{Exp}_{\tilde{x}^s}^{-1}(x_t^{s+1})\|^2\right]$$

$$= \mathbb{E}\left[f(x_t^{s+1}) - \left(\eta - \frac{c_{t+1}\eta}{\beta}\right)\|\nabla f(x_t^{s+1})\|^2\right]$$

$$+ \left(\frac{L\eta^2}{2} + c_{t+1}\zeta\eta^2\right)\mathbb{E}\left[\|v_t^{s+1}\|^2\right]$$

$$+ (c_{t+1} + c_{t+1}\eta\beta)\mathbb{E}\left[\|\mathrm{Exp}_{\tilde{x}^s}^{-1}(x_t^{s+1})\|^2\right] \tag{8}$$

It remains to bound $\mathbb{E}\left[\|v_t^{s+1}\|^2\right]$. Denoting $\Delta_t^{s+1} = \nabla f_{i_t}(x_t^{s+1}) - \Gamma_{\tilde{x}^s}^{x_t^{s+1}}\nabla f_{i_t}(\tilde{x}^s)$, we have $\mathbb{E}[\Delta_t^{s+1}] = \nabla f(x_t^{s+1}) - \Gamma_{\tilde{x}^s}^{x_t^{s+1}}\nabla f(\tilde{x}^s)$, and thus

$$\mathbb{E}\left[\|v_t^{s+1}\|^2\right] = \mathbb{E}\left[\|\Delta_t^{s+1} + \Gamma_{\tilde{x}^s}^{x_t^{s+1}}\nabla f(\tilde{x}^s)\|^2\right]$$

$$= \mathbb{E}\left[\|\Delta_t^{s+1} - \mathbb{E}[\Delta_t^{s+1}] + \nabla f(x_t^{s+1})\|^2\right]$$

$$\leq 2\mathbb{E}[\|\Delta_t^{s+1} - \mathbb{E}[\Delta_t^{s+1}]\|^2] + 2\mathbb{E}[\|\nabla f(x_t^{s+1})\|^2]$$

$$\leq 2\mathbb{E}[\|\Delta_t^{s+1}\|^2] + 2\mathbb{E}[\|\nabla f(x_t^{s+1})\|^2]$$

$$\leq 2L^2\mathbb{E}[\|\mathrm{Exp}_{\tilde{x}^s}^{-1}(x_t^{s+1})\|^2] + 2\mathbb{E}[\|\nabla f(x_t^{s+1})\|^2] \tag{9}$$

where the first inequality is due to $\|a + b\|^2 \leq 2\|a\|^2 + 2\|b\|^2$, the second due to $\mathbb{E}\|\xi - \mathbb{E}\xi\|^2 = \mathbb{E}\|\xi\|^2 - \|\mathbb{E}\xi\|^2 \leq \mathbb{E}\|\xi\|^2$ for any random vector $\xi$ in any tangent space, the third due to $L$-g-smooth assumption. Substituting Equation (9) into Equation (8) we get

$$
\begin{aligned}
R_{t+1}^{s+1} \leq{} & \mathbb{E}\left[ f(x_t^{s+1}) - \left( \eta - \frac{c_{t+1}\eta}{\beta} - L\eta^2 - 2c_{t+1}\zeta\eta^2 \right) \|\nabla f(x_t^{s+1})\|^2 \right] \\
& + \left( c_{t+1}\left(1 + \beta\eta + 2\zeta L^2\eta^2\right) + L^3\eta^2 \right) \mathbb{E}\left[ \|\mathrm{Exp}_{\tilde{x}^s}^{-1}(x_t^{s+1})\|^2 \right] \\
={} & R_t^{s+1} - \left( \eta - \frac{c_{t+1}\eta}{\beta} - L\eta^2 - 2c_{t+1}\zeta\eta^2 \right) \mathbb{E}\left[ \|\nabla f(x_t^{s+1})\|^2 \right]
\end{aligned}
\tag{10}
$$

Rearranging terms completes the proof. $\qquad\square$

**Theorem 6.** With assumptions as in Lemma 2, let $c_m = 0, \eta > 0, \beta > 0$, and $c_t = c_{t+1}\left(1 + \beta\eta + 2\zeta L^2\eta^2\right) + L^3\eta^2$ such that $\delta(t) > 0$ for $0 \leq t \leq m-1$. Define the quantity $\delta_n := \min_t \delta(t)$, and let $T = mS$. Then for the output $x_a$ from Option II we have

$$
\mathbb{E}[\|\nabla f(x_a)\|^2] \leq \frac{f(x^0) - f(x^*)}{T\delta_n}
$$

*Proof.* Using Lemma 2 and telescoping the sum, we obtain

$$
\sum_{t=0}^{m-1} \mathbb{E}[\|\nabla f(x_t^{s+1})\|^2] \leq \frac{R_0^{s+1} - R_m^{s+1}}{\delta_n}
$$

Since $c_m = 0$ and $x_0^{s+1} = \tilde{x}^s$, we thus have

$$
\sum_{t=0}^{m-1} \mathbb{E}[\|\nabla f(x_t^{s+1})\|^2] \leq \frac{\mathbb{E}[f(\tilde{x}^s) - f(\tilde{x}^{s+1})]}{\delta_n},
\tag{11}
$$

Now sum over all epochs to obtain

$$
\frac{1}{T} \sum_{s=0}^{S-1} \sum_{t=0}^{m-1} \mathbb{E}[\|\nabla f(x_t^{s+1})\|^2] \leq \frac{f(\tilde{x}^0) - f(x^*)}{T\delta_n}
\tag{12}
$$

Note the definition of $x_a$ implies that the left hand side of (12) is exactly $\mathbb{E}[\|\nabla f(x_a)\|^2]$. $\qquad\square$

**Theorem 2.** Assume in (1) each $f_i$ is $L$-g-smooth, the sectional curvature in $\mathcal{X}$ is lower bounded by $\kappa_{\min}$, and we run Algorithm 1 with Option II. Then there exist universal constants $\mu_0 \in (0,1), \nu > 0$ such that if we set $\eta = \mu_0/(Ln^{\alpha_1}\zeta^{\alpha_2})$ ($0 < \alpha_1 \leq 1$ and $0 \leq \alpha_2 \leq 2$), $m = \lfloor n^{3\alpha_1/2}/(3\mu_0\zeta^{1-2\alpha_2}) \rfloor$ and $T = mS$, we have

$$
\mathbb{E}[\|\nabla f(x_a)\|^2] \leq \frac{Ln^{\alpha_1}\zeta^{\alpha_2}[f(x^0) - f(x^*)]}{T\nu},
$$

where $x^*$ is an optimal solution to the problem in (1).

*Proof.* Let $\beta = L\zeta^{1-\alpha_2}/n^{\alpha_1/2}$. From the recurrence relation $c_t = c_{t+1}\left(1 + \beta\eta + 2\zeta L^2\eta^2\right) + L^3\eta^2$ and $c_m = 0$ we have

$$
c_0 = \frac{\mu_0^2 L}{n^{2\alpha_1}\zeta^{2\alpha_2}} \frac{(1+\theta)^m - 1}{\theta},
$$

where

$$
\theta = \eta\beta + 2\zeta\eta^2 L^2 = \frac{\mu_0\zeta^{1-2\alpha_2}}{n^{3\alpha_1/2}} + \frac{2\mu_0^2\zeta^{1-2\alpha_2}}{n^{2\alpha_1}} \in \left( \frac{\mu_0\zeta^{1-2\alpha_2}}{n^{3\alpha_1/2}}, \frac{3\mu_0\zeta^{1-2\alpha_2}}{n^{3\alpha_1/2}} \right).
$$

Notice that $\theta < 1/m$ so that $(1+\theta)^m < e$. We can thus bound $c_0$ by

$$
c_0 \leq \frac{\mu_0 L}{n^{\alpha_1/2}\zeta}(e-1)
$$

and in turn bound $\delta_n$ by

$$\delta_n = \min_t \left( \eta - \frac{c_{t+1}\eta}{\beta} - \eta^2 L - 2c_{t+1}\zeta\eta^2 \right)$$

$$\geq \left( \eta - \frac{c_0\eta}{\beta} - \eta^2 L - 2c_0\zeta\eta^2 \right)$$

$$\geq \eta \left( 1 - \frac{\mu_0(e-1)}{\zeta^{2-\alpha_2}} - \frac{\mu_0}{n^{\alpha_1}\zeta^{\alpha_2}} - \frac{2\mu_0^2(e-1)}{n^{3\alpha_1/2}\zeta^{\alpha_2}} \right)$$

$$\geq \frac{\nu}{Ln^{\alpha_1}\zeta^{\alpha_2}}$$

where the last inequality holds for small enough $\mu_0$, as $\zeta, n \geq 1$. For example, it holds for $\mu_0 = 1/10, \nu = 1/20$. Substituting the above bound in Theorem 6 concludes the proof. $\qquad \square$

**Corollary 2.** With assumptions and parameters in Theorem 2, choosing $\alpha_1 = 2/3$, the IFO complexity for achieving an $\epsilon$-accurate solution is:

$$\text{IFO calls} = \begin{cases} O\left(n + (n^{2/3}\zeta^{1-\alpha_2}/\epsilon)\right), & \text{if } \alpha_2 \leq 1/2, \\ O\left(n\zeta^{2\alpha_2-1} + (n^{2/3}\zeta^{\alpha_2}/\epsilon)\right), & \text{if } \alpha_2 > 1/2. \end{cases}$$

*Proof.* Note that to reach an $\epsilon$-accurate solution, $O(n^{\alpha_1}\zeta^{\alpha_2}/(m\epsilon)) = O(1 + n^{-1/3}\zeta^{1-\alpha_2}/\epsilon)$ epochs are required. On the other hand, one epoch takes $O\left(n(1 + \zeta^{2\alpha_2-1})\right)$ IFO calls. Thus the total amount of IFO calls is $O\left(n(1 + \zeta^{2\alpha_2-1})(1 + n^{-1/3}\zeta^{1-\alpha_2}/\epsilon)\right)$. Simplify to get the stated result. $\qquad \square$

**Theorem 3.** Suppose that in addition to the assumptions in Theorem 2, $f$ is $\tau$-gradient dominated. Then there exist universal constants $\mu_0 \in (0,1), \nu > 0$ such that if we run Algorithm 2 with $\eta = \mu_0/(Ln^{2/3}\zeta^{1/2}), m = \lfloor n/(3\mu_0) \rfloor, S = \lceil (6 + \frac{18\mu_0}{n-3})L\tau\zeta^{1/2}\mu_0/(\nu n^{1/3}) \rceil$, we have

$$\mathbb{E}[\|\nabla f(x^K)\|^2] \leq 2^{-K}\|\nabla f(x^0)\|^2,$$
$$\mathbb{E}[f(x^K) - f(x^*)] \leq 2^{-K}[f(x^0) - f(x^*)].$$

*Proof.* Apply Theorem 2. Observe that for each run of Algorithm 1 with Option II we now have $T = mS \geq 2L\tau n^{2/3}\zeta^{1/2}/\nu$, which implies

$$\frac{1}{\tau}\mathbb{E}[f(x^{k+1}) - f(x^*)] \leq \mathbb{E}[\|\nabla f(x^{k+1})\|^2] \leq \frac{1}{2\tau}\mathbb{E}[f(x^k) - f(x^*)] \leq \frac{1}{2}\mathbb{E}[\|\nabla f(x^k)\|^2]$$

The theorem follows by recursive application of the above inequality. $\qquad \square$

**Corollary 4.** With Algorithm 2 and the parameters in Theorem 3, the IFO complexity to compute an $\epsilon$-accurate solution for gradient dominated function $f$ is $O((n + L\tau\zeta^{1/2}n^{2/3})\log(1/\epsilon))$.

*Proof.* We need $O((n+m)S) = O(n + L\tau\zeta^{1/2}n^{2/3})$ IFO calls in a run of Algorithm 1 to double the accuracy, thus in Algorithm 2, $K = O(\log(1/\epsilon))$ runs are needed to reach $\epsilon$-accuracy. $\qquad \square$

**Corollary 5.** With Algorithm 2 and the parameters in Theorem 3, the IFO complexity to compute an $\epsilon$-accurate solution for a $\mu$-strongly g-convex function $f$ is $O((n + \mu^{-1}L\zeta^{1/2}n^{2/3})\log(1/\epsilon))$.

*Proof.* Assume $x^*$ is the minimizer of $f$ and $f$ is $\mu$-strongly g-convex, then we have

$$f(x^*) = \min_y f(y)$$

$$\geq \min_y f(x) + \langle \nabla f(x), \text{Exp}_x^{-1}(y) \rangle + \frac{\mu}{2}\|\text{Exp}_x^{-1}(y)\|^2$$

$$= f(x) - \frac{1}{2\mu}\|\nabla f(x)\|^2 + \min_y \frac{1}{2\mu}\|\nabla f(x) + \mu\text{Exp}_x^{-1}(y)\|^2$$

$$\geq f(x) - \frac{1}{2\mu}\|\nabla f(x)\|^2$$

where we get the first inequality by strong g-convexity, the second equality by completing the squares, and the second inequality by choosing $y = \text{Exp}_x\left(-\frac{1}{\mu}\nabla f(x)\right)$. Thus $f(x)$ is $(1/(2\mu))$-gradient dominated, and choosing $\tau = 1/(2\mu)$ in Corollary 4 concludes the proof. $\qquad \square$

# C Proof for Section 4.1

**Theorem 4.** Suppose $A$ has eigenvalues $\lambda_1 > \lambda_2 \geq \cdots \geq \lambda_d$ and $\delta = \lambda_1 - \lambda_2$, and $x^0$ is drawn uniformly randomly on the hypersphere. Then with probability $1 - p$, $x^0$ falls in a Riemannian ball of a global optimum of the objective function, within which the objective function is $O(\frac{d}{p^2\delta})$-gradient dominated.

*Proof.* We write $x$ in the basis of $A$'s eigenvectors $\{v_i\}_{i=1}^d$ with corresponding eigenvalues $\lambda_1 > \lambda_2 \geq \cdots \geq \lambda_d$, i.e. $x = \sum_{i=1}^d \alpha_i v_i$. Thus $Ax = \sum_{i=1}^d \alpha_i \lambda_i v_i$ and $f(x) = -\sum_{i=1}^d \alpha_i^2 \lambda_i$. The Riemannian gradient of $f(x)$ is $P_x \nabla f(x) = -2(I - xx^\top)Ax = -2(Ax + f(x)x) = -2\sum_{i=1}^d \alpha_i(\lambda_i - \sum_{j=1}^d \alpha_j^2 \lambda_j)v_i$ (see [1, Example 3.6.1]). Now consider a Riemannian ball on the hypersphere defined by $\mathcal{B}_\epsilon \triangleq \{x : x \in \mathbb{S}^{d-1}, \alpha_1 \geq \epsilon\}$, note that the center of $\mathcal{B}_\epsilon$ is the first eigenvector. We apply a case by case argument with respect to $f(x) - f(x^*)$. If $f(x) - f(x^*) \geq \frac{\delta}{2}$, we can lower bound the gradient by

$$\frac{1}{4}\|P_x \nabla f(x)\|^2 = \sum_{i=1}^d \alpha_i^2 \left(\lambda_i - \sum_{j=1}^d \alpha_j^2 \lambda_j\right)^2 \geq \alpha_1^2\left(\lambda_1 - \sum_{j=1}^d \alpha_j^2 \lambda_j\right)^2 = \alpha_1^2\left(f(x) - f(x^*)\right)^2$$
$$\geq \frac{1}{2}\alpha_1^2 \delta(f(x) - f(x^*)) \geq \frac{1}{2}\epsilon^2 \delta(f(x) - f(x^*))$$

The last equality follows from the fact that $f(x^*) = -\lambda_1$ and $f(x) = -\sum_{i=1}^d \alpha_i^2 \lambda_i$. On the other hand, if $f(x) - f(x^*) < \frac{\delta}{2}$, for $i = 2, \ldots, d$, since $-\lambda_i - f(x^*) \geq \delta$, we have $-\lambda_i - f(x) > \frac{1}{2}(-\lambda_i - f(x^*)) \geq \delta/2$. We can, again, lower bound the gradient by

$$\|P_x \nabla f(x)\|^2 = 4\sum_{i=1}^d \alpha_i^2\left(\lambda_i - \sum_{j=1}^d \alpha_j^2 \lambda_j\right)^2 \geq 4\sum_{i=2}^d \alpha_i^2\left(\lambda_i - \sum_{j=1}^d \alpha_j^2 \lambda_j\right)^2$$
$$\geq \sum_{i=2}^d \alpha_i^2\left(\lambda_1 - \lambda_i\right)^2 \geq \delta\sum_{i=2}^d \alpha_i^2\left(\lambda_1 - \lambda_i\right) = \delta(f(x) - f(x^*))$$

Combining the two cases, we have that within $\mathcal{B}_\epsilon$ the objective function (5) is $\max\{\frac{1}{2\epsilon^2\delta}, \frac{1}{\delta}\}$-gradient dominated. Finally, observe that if $x^0$ is chosen uniformly at random on $\mathbb{S}^{d-1}$, then with probability at least $1 - p$, $\alpha_1^2 = \Omega(\frac{p^2}{d})$, i.e. there exists some constant $c > 0$ such that $\frac{1}{\epsilon^2} \leq \frac{cd}{p^2}$. □