[Reviews · NeurIPS 2016]

Reviewer 1

Summary

This proposes a variant of the SVRG method that applies to optimization over Riemann manifolds. The work analyzes the convergence rate method under a general non-convex optimization setting, and under a much stronger growth condition on the gradient.

Qualitative Assessment

The work is a nice combination of existing works. I'm not an expert on the Riemann side of things and although some effort is made in this direction it still isn't really written for such readers. Thus, I'm giving this review with very low confidence, and hope that another reviewer can comment on this part. Unless I'm missing something, the discussion after Corollary 1 indicates that the RSVRG has at best the same speed as the regular SVRG. Am I misintepreting this? Note that prior to Reddi et al.'s work, SVRG was analyzed without strong convexity in 2014 by Gong and Ye. Theorem 5 is a nice result, although the experiments are underwhelming. Doesn't this indicate we can get similar performance with The paper would be strengthened if the discussion included a list of other problems where the Riemannian geometry is known to help.

Confidence in this Review

1-Less confident (might not have understood significant parts)


Reviewer 2

Summary

The paper generalizes the popular SVRG from the vector space to Riemannian manifolds, and provides the convergence analysis for geodesically strongly convex, convex and nonconvex functions.

Qualitative Assessment

Given the popularity and practical value of SVRG, extending it to Riemannian manifolds is an important step. However, the development and analysis of RSVRG for geodesically convex functions may not make sense, because compact Riemannian manifolds with finite volume only admit constant convex functions (BISHOP & O'NEILL, Manifolds of negative curvature, 1969). The proof of theorem 4 is not clear.

Confidence in this Review

2-Confident (read it all; understood it all reasonably well)


Reviewer 3

Summary

This paper considers optimizing a finite-sum cost function on Riemannian manifold. The Euclidean stochastic variance reduced gradient method is generalized to the Riemannian setting using the notion of exponential mapping and parallel translation. Non-asymtotic convergence analysese are given for both geodesic-convex and geodesic-nonconvex funcions. PCA problem and Riemannian centroid problem are used to show the effectiveness and effeciency of the proposed method.

Qualitative Assessment

This paper is well written. The analysis results are interesting. The experimental results are convincing. One comment about the proof of Theorem 1: The first inequality after line 351 is not clear. It seems that the authors assume that \Gamma_{x^*}^{x_t^{s+1}} = \Gamma_{\tilde{x}^s}^{{x_t^{s+1}}} \circ \Gamma_{x^*}^{\tilde{x}^s}. In other words, the authors assume that the parallel translation from x_1 to x_3 is equal to the composition of the parallel translation from x_1 to x_2 and from x_2 to x_3. This is not true in general. The parallel translation is dependent on path in general. I think this problem can be fixed, but the coefficient on the inequality should be different. Another comment: the curvature of manifold affects the convergence rate. It would be nice if experiments, which show the influence of the curvature of manifold, are given.

Confidence in this Review

2-Confident (read it all; understood it all reasonably well)


Reviewer 4

Summary

The authors consider the problem of finite sum minimization, in the setting where the loss functions as well as the constraints may be non-convex. Specifically, they concentrate on the case where decision points are constrained to be on a Riemannian manifold, and the losses are g-convex. Then they devise an SVRG variant for this setting and show that for smooth and g-strongly convex objectives this variant has a runtime complexity in which the dimension $n$ and the condition number $L/\mu$ are additive ( and the dependence on the accuracy $\epsilon$ is logarithmic). The authors additionally analyze SVRG variants for non g-convex objectives (where we seek to converge to a stationary point) and for gradient dominated objectives.

Qualitative Assessment

The investigation of optimization problems over Riemmanian manifolds is a very interesting line of research. Concretely, the extension of SVRG to this setting is an important contribution in this line, both theoretically and practically. While the RSVRG algorithm is quite natural, its analysis, especially Lemma 2, seems quite original. Nevertheless, the proofs of Lemma 2, and of Theorem 1, lack quite a lot of details, which make it impossible to validate. These missing details are mainly related to mathematical relations regarding Riemannian manifolds. Concretely: Missing details in the Proof of Lemma 2: - The second equality in the line below 153 is unclear. - The part discussing the optimality of the solution appearing in Equation (7) is unclear (a very vague explanation) -The calculation of $\lambda$ is unclear Missing details in the Proof of Theorem 1: - The inequality below 351 is unclear, where does the parallel transport vanish? - You seem to use the linearity of the parallel transport , but this is not mentioned anywhere, is this operator indeed linear? Finally, the paper does not discuss the differences between manifolds with negative/positive curvature. It is worthwhile to give some intuition and discuss the differences between these two cases (beyond the very technical discussion that is conducted) To conclude, this work is an interesting and important progress (both theoretically and practically). However, the current manuscript does not enable to validate its correctness.

Confidence in this Review

2-Confident (read it all; understood it all reasonably well)


Reviewer 5

Summary

This paper proposes a stochastic variance reduced gradient (SVRG) method for geodesically smooth functions on riemannian manifolds. They show better convergence complexities, depending on sectional curvature, for both geodesically convex and non convex functions. Experiments are presented for PCA and computing the riemannian centroid.

Qualitative Assessment

The paper is well written and reasonably clear for the reader who already know optimization on riemannian manifolds and svrg literature. Though the paper is incremental, they gives better analyses for convergence complexities. Especially, the dependency on sectional curvature is interesting. The experiments is rather weak. For the riemannian centroid problem, I want the author to compare the method to non-riemannian methods in order to show the advantage of it.

Confidence in this Review

2-Confident (read it all; understood it all reasonably well)


Reviewer 6

Summary

This paper develops RSVRG, a variance-reduced stochastic gradient method for minimizing finite sums of functions on Riemannian manifolds. The paper establishes that the approach can converge linearly, even for geodesically nonconvex functions, and that the complexity of the method depends on the geometry of the manifold. The convergence results are nonasymptotic, and the complexity results use the IFO paradigm. The paper provides experiments for two applications. One application is PCA and the other is computing the Riemannian centroid of symmetric positive definite matrices. Through these experiments, the paper illustrates the benefit of variance reduction.

Qualitative Assessment

This paper addresses a topic that was proposed as future work in [32], and by design, much of its contents are adaptations of results found in [14] and [21]. Thus, the analysis is incremental. Nonetheless, the paper contributes a manifoldized algorithm and bridges a gap between convex optimization and Riemannian optimization. Furthermore, the paper provides a useful lemma to analyze Riemannian methods, which can have a lasting impact. My main concern with this paper is its presentation. The paper is elusive on details, which undermines its technical quality, omits an important reference, and suffers from language issues. In Theorem 4, the constants mu1 and nu1 should be defined (are they the same as the constants in [21]?). For the PCA application, the matrix Z should be related to the problem (are the vectors z the columns of Z as in [25]?). For both applications, it should be mentioned how the accuracy is measured (accuracy is the ordinate of most of the plots). For the centroid application, the parameters N and Q should be defined (is N=n?). The paper should expound on the exponential and parallel transport mappings (the most important parts of RSVRG) for each of the applications: granted, for the PCA application, the paper cites an entire textbook for the mappings, but a citation like this is more pleasant with page numbers (I could not find the relevant results). The reference for the IFO framework should be included; the reference is "A Lower Bound for the Optimization of Finite Sums" by Alekh Agarwal and Léon Bottou. Finally, the paper would benefit from a good proof read. Here are some of the language issues: Line 48: "developing fast stochastic for it". Maybe this should read "developing fast stochastic methods for it". Line 56-57: "we generalize convergence analysis". This should read "we generalize the convergence analysis"; there are many missing "the's" in the paper. Line 59: "method this is provably". Perhaps "this" should be replaced by "that", or the sentence should be reworded. Line 130: "takes and i". Not "and" but "an". Line 155: "an fi minimize all". Not "minimize" but "minimizes". Theorem 2: The first sentence is incomplete. Maybe "If" should be replaced by "Suppose" (as in the restatement of the theorem in the supplementary material). Here are some other issues: In the experiments for PCA, the methodology for estimating the number of accuracy-doubling epochs should be described. This number becomes less for RSVRG than for VR-PCA as the eigengap decreases, and it should be mentioned whether the number of IFO calls follows this tendency for a given accuracy (the comparison is provided only for a single, relatively large delta). It should be explained why the IFO complexity provides a fair comparison (RSVRG requires evaluating not only gradients, but also exponential and parallel transport mappings - are the extra computations worth it?). More issues: Equation 8: The index i on each z is missing. Line 435 (in the supplementary material): "Corollary ??".

Confidence in this Review

2-Confident (read it all; understood it all reasonably well)